# Targeting the Autonomic Nervous System for Risk Stratification, Outcome Prediction and Neuromodulation in Ischemic Stroke

**DOI:** 10.3390/ijms22052357

**Published:** 2021-02-26

**Authors:** Angelica Carandina, Giulia Lazzeri, Davide Villa, Alessio Di Fonzo, Sara Bonato, Nicola Montano, Eleonora Tobaldini

**Affiliations:** 1Department of Internal Medicine, Fondazione IRCCS Ca’ Granda, Ospedale Maggiore Policlinico, 20122 Milan, Italy; angelica.carandina@policlinico.mi.it (A.C.); eleonora.tobaldini@unimi.it (E.T.); 2Neurology Unit, Fondazione IRCCS Ca’ Granda, Ospedale Maggiore Policlinico, 20122 Milan, Italy; giulia.lazzeri@unimi.it (G.L.); alessio.difonzo@policlinico.mi.it (A.D.F.); 3Dino Ferrari Centre, Neuroscience Section, Department of Pathophysiology and Transplantation, University of Milan, 20122 Milan, Italy; davide.villa@unimi.it (D.V.); sara.bonato@policlinico.mi.it (S.B.); 4Stroke Unit, Fondazione IRCCS Ca’ Granda, Ospedale Maggiore Policlinico, 20122 Milan, Italy; 5Department of Clinical Sciences and Community Health, University of Milan, 20122 Milan, Italy

**Keywords:** stroke, autonomic nervous system, heart rate variability, vagus nerve stimulation, risk stratification

## Abstract

Ischemic stroke is a worldwide major cause of mortality and disability and has high costs in terms of health-related quality of life and expectancy as well as of social healthcare resources. In recent years, starting from the bidirectional relationship between autonomic nervous system (ANS) dysfunction and acute ischemic stroke (AIS), researchers have identified prognostic factors for risk stratification, prognosis of mid-term outcomes and response to recanalization therapy. In particular, the evaluation of the ANS function through the analysis of heart rate variability (HRV) appears to be a promising non-invasive and reliable tool for the management of patients with AIS. Furthermore, preclinical molecular studies on the pathophysiological mechanisms underlying the onset and progression of stroke damage have shown an extensive overlap with the activity of the vagus nerve. Evidence from the application of vagus nerve stimulation (VNS) on animal models of AIS and on patients with chronic ischemic stroke has highlighted the surprising therapeutic possibilities of neuromodulation. Preclinical molecular studies highlighted that the neuroprotective action of VNS results from anti-inflammatory, antioxidant and antiapoptotic mechanisms mediated by α7 nicotinic acetylcholine receptor. Given the proven safety of non-invasive VNS in the subacute phase, the ease of its use and its possible beneficial effect in hemorrhagic stroke as well, human studies with transcutaneous VNS should be less challenging than protocols that involve invasive VNS and could be the proof of concept that neuromodulation represents the very first therapeutic approach in the ultra-early management of stroke.

## 1. Introduction

Cerebrovascular disease ranks as one of the leading causes of disability and mortality worldwide [1,2]. Acute ischemic stroke (AIS), induced by abnormal cerebral perfusion due to sudden rupture or occlusion of cerebral vessels, is the most common type of stroke, accounting for 70% of all stroke cases [1]. In the general population, acute ischemic stroke has been found to be more frequent in men than in women [3]. However, women experience more severe strokes and have longer hospitalizations than men, resulting in higher percentages of permanent disability and mortality [4,5].

Autonomic nervous system (ANS) dysfunction and ischemic stroke have an intricate and deep interconnection. The disruption of autonomic regulatory pathways may determine clinical complications during both the acute and chronic phases of stroke, leading to a worse prognosis [6]. On the other hand, an ANS imbalance can contribute to the creation of a pre-pathological milieu of composite risk factors, such as hypertension, diabetes or atrial fibrillation, and alterations of endothelial homeostasis in favor of pro-thrombotic/pro-inflammatory state, eventually leading to increased risk of AIS.

The nature of the cause–effect relation between ANS dysfunction and stroke is currently not entirely established. Nevertheless, even a partial unravelment of such an ambivalent role played by ANS, linking both risk factors and clinical features of stroke, may be essential, constituting a premise for the better therapeutic management of primary and secondary prevention of AIS in clinical practice. In this context, we here review the emerging evidence linking the ANS to the pathological process of stroke, the effects of ANS dysregulation after the ischemic event and the potential role of heart variability (HRV) analysis, a non-invasive method to assess ANS function, in the diagnostic workup of stroke patients. In addition, we will discuss the promising role of invasive and non-invasive vagal stimulation as a therapeutic approach in acute and chronic phases of stroke.

## 2. ANS Dysfunction and Ischemic Stroke: Running in Circles

### 2.1. ANS Dysfunction Can Increase the Risk of Ischemic Stroke

The 2009 standardized case–control INTERSTROKE study in 22 countries worldwide found that hypertension, current smoking, diabetes, abdominal obesity, poor diet and physical inactivity accounted for more than 80% of the global risk of both ischemic and hemorrhagic stroke; other risk factors included excessive alcohol consumption, dyslipidemia, cardiac causes (atrial fibrillation or flutter, previous myocardial infarction, rheumatic valvular heart disease and prosthetic heart valve) and psychosocial stress/depression [7]. The majority of the above-mentioned stroke risk factors are more common in males but, when considering only stroke patients, they are shown to increase stroke risk more in women than in men [8]. Interestingly, many of these predisposing factors increase not only the risk of ischemic stroke but also of autonomic dysfunction. However, autonomic dysfunction, regardless of its primary cause, can itself be associated with increased risk of ischemic stroke. For instance, autonomic imbalance, with a shift towards sympathetic predominance, may lead to the development of atrial fibrillation (AF) or hypertension. [9,10] (Figure 1).

Heart rate variability (HRV) is a non-invasive method to explore cardiac ANS function; thus, it is not surprising that the majority of the scientific literature employs this tool to predict and monitor cardiovascular disease and associated risk factors. HRV is based on the identification of oscillatory rhythmic components: the low-frequency oscillation (LF—ranging from 0.04 to 0.15 Hz) and the high-frequency oscillation (HF—ranging from 0.15 to 0.4 Hz and synchronous with respiration). The power of the HF band is determined by parasympathetic modulation and the power of the LF band reflects sympathetic modulation and cardiac autonomic outflows by baroreflexes. Changes in sympatho-vagal balance can be detected by the LF/HF ratio [11,12]. Conditions characterized by a reduction in vagal modulation and in global HRV do represent recognized prognostic risk factors for adverse cardiovascular events as well as cardiac mortality [13].

Different reports suggest that reduced HRV, assessed during both daytime and night-time recordings, is strongly associated with incident stroke in apparently healthy subjects with no history of cardiovascular disease or stroke [14] and in the general population [15]. Similar results have been obtained by autonomic assessment in cohorts of high-risk subjects, such as hypertensive [16] and diabetic patients [17,18,19].

Hypertension is the most prevalent risk factor of ischemic stroke [20]. An association between autonomic dysfunction and hypertension can be partially explained by the evidence of reduced baroreflex sensitivity [21], as well as by an increase in sympatho-sympathetic positive feedback reflexes [22]. HRV is known to be reduced in patients with systemic hypertension [23,24,25], especially in the presence of uncontrolled blood pressure [26]. Among normotensive men, lower HRV may be even associated with a higher risk of incident hypertension [27].

AF is a very common cardiac arrhythmia, affecting approximately 1–3% of the general population and whose prevalence rises with age [28]. Its association with ischemic stroke is more complex than a straightforward cause–effect dependence, as widely discussed by Kamel et. al. [29]. Overall, almost one third of ischemic stroke patients have either a history of AF or newly diagnosed AF after stroke [30]. AF is associated with a 1.5–1.9-times increased risk of mortality and 2–5-times increased risk of stroke, transient ischemic attack (TIA) or systemic embolism [31]. Cardiac emboli can be of various sizes, but those originating in the cardiac chambers are frequently large and therefore especially likely to cause severe stroke, disability and death. Left atrial appendage has been reported as the main source of AF-related thromboemboli (89% of cases) [32]. Cardioembolic infarction is generally associated with a poor outcome, presenting the highest in-hospital mortality rate among stroke subtypes, due to an increased risk of early embolic recurrence [33]. ANS can induce relevant electrophysiological changes in atrial cardiomyocytes, playing a fundamental role in both the initiation and maintenance of AF [34]; in particular, before an episode of AF, the occurrence of a primary increase in adrenergic tone followed by an abrupt shift toward vagal predominance can commonly be observed [35]. An interesting study analyzing HRV in a cohort of 11,715 patients, with a mean follow-up time of almost 20 years, revealed that lower HRV was significantly associated with a higher AF risk [36]. Similar results were obtained by analyzing pulse rate variability (PRV), measured by a blood pressure monitor, with AF and cerebrovascular risk rising with increasing pulse irregularity [37]. Furthermore, autonomic dysfunction, expressed as a higher LF/HF ratio, has been shown to predict silent AF episodes in diabetic patients [38]. In subjects with permanent AF, classically excluded from standard analysis of HRV, multiscale entropy (MSE) as a measure of the irregularity of the ventricular response interval has demonstrated potentiality in predicting ischemic stroke in a study by Watanabe et al. [39]. Therefore, ANS imbalance may contribute not only to the development of AF but also to its putative pathophysiological substrate, atrial cardiopathy, with relevant implications in the pathogenesis of cryptogenic stroke [40].

Large vessel atherosclerosis, involving both extracranial and intracranial arteries, is reported to be the cause of approximately 15% of ischemic strokes [41]. The pathophysiological process leading to the accumulation of fatty and fibrous material in the intima of arterial vessels is still not completely explained [42]. However, the first hypothesis about an essential and possibly initiating role played by inflammation in the development of atherosclerotic plaques dates back to the mid-19th century, with Virchow’s research [43]. This inflammatory process may be influenced by autonomic terminals on arteries, leading to endothelial dysfunction [44]. Such a potential role of ANS in the pathogenesis of atherosclerosis has been demonstrated in animal models, with a marked reduction in the progression of atherosclerosis after sympathectomy [45]. On the other hand, apart from being a possible co-player in the atherosclerotic process, ANS dysfunction can eventually be a consequence of carotid atherosclerosis, interfering with carotid baro- and chemo-receptor function, culminating in sympathetic predominance and reduced vagal tone [46,47]. Interestingly, the strict link between ANS dysfunction, inflammation and atherosclerosis appears to be ascertainable in vivo as well, as shown by Ulleryd et al. [48] and Rupprecht et al. [49].

Recent reports suggest that up to one third of stroke patients suffer from diabetes mellitus (DM) [50]. The ANS is involved in the regulation of the homeostasis of glucose levels; thus, autonomic dysfunction may precede and even be a predisposing factor to the development of DM. Pancreatic islets receive both sympathetic and vagal innervation and the hypothalamus is the key regulator of glucose metabolism; coherently, animal models of both central and peripheral ANS lesions have shown a secondary effect on insulin secretion. Therefore, sympathetic overactivity can be considered one of the primary causes of insulin resistance, eventually leading to the development of DM, as elegantly explained by Frontoni et al. [51]. On the other hand, diabetic autonomic neuropathy (DAN), a very well-known complication of DM [52], is an independent risk factor for ischemic stroke [19,53].

Reduced HRV has also been shown to be inversely associated with carotid stenosis severity [49], suggesting that it may be a useful tool for detecting high-risk patients who may benefit from revascularization or surgical procedures. Other important risk factors for AIS and cardiovascular disease have also been linked to hypercholesterolemia [54]. In the latter case, long-term statin therapy has been shown to partially revert the reduction in HRV [55].

Finally, it would be helpful to mention that ischemic stroke may be the initial manifestation of hematologic diseases and also that both inherited (such as factor V Leiden and prothrombin *G20210A* mutations, deficiencies of protein C, protein S and anti-thrombin) and acquired hypercoagulable disorders (notably anti-phospholipid syndrome and neoplastic thrombophilia) may be a commonly unrecognized cause of ischemic stroke, particularly in young patients [56,57,58]. Moreover, hematologic disorders, especially myeloproliferative neoplasms, have also been associated with a higher risk of AIS [59]. Unfortunately, literature about autonomic analysis in these subsets of patients is scarce. However, reduced HRV has been demonstrated in specific cohorts, such as treatment-naïve unselected cancer patients [60], and autonomic dysfunction in anti-phospholipid syndrome has been shown to closely correlate with thrombophilic state [61]. As for ANS function, both in vitro and in vivo studies have demonstrated that increased sympathetic activity may trigger a hypercoagulable state per se, either by platelet activation [62], or by local hemostatic activity increasing vascular resistance [63]. Therefore, autonomic imbalance towards a sympathetic shift may have a relevant role not only as a chronic risk factor for AIS but also in the hyperacute phase of clot formation. ANS dysfunction in cerebral ischemia due to hematological diseases is a clinical aspect that has not been thoroughly assessed and that could also be of prognostic and outcome interest.

Taking evidence into account, HRV analysis may provide an accessible, fast and accurate risk stratification tool for both healthy and high-risk patients who may benefit from either educational, pharmacological or even interventional prevention approaches, since many of the main risk factors for stroke are modifiable [64].

### 2.2. ANS Dysfunction after Ischemic Stroke

Cardiovascular dysfunction due to central nervous system lesions has been investigated extensively during the last few decades [65,66]. Apart from classic clinical features such as hemiplegia or aphasia, a wide spectrum of signs and symptoms can insidiously occur when ANS is involved with peculiar patterns, in relation to the site and the extension of brain lesions (Figure 2). Cardiovascular abnormalities typically encountered during the acute phase of ischemic stroke include serious arrhythmias, conduction defects and repolarization anomalies [67], and even myocardial damage [68]. Additionally, most patients suffering from ischemic stroke reveal high values of blood pressure [69], a compensatory reaction to reduced brain blood flow, but eventually leading to poorer outcomes [70,71,72]. Sympathetic overactivity, suggested by high levels of serum catecholamines, may also lead to hyperglycemia [73], another factor associated with worse outcomes [74], and even to a higher risk of infection in the acute phase [75,76]. Relative overactivity of both parasympathetic and sympathetic branches has been involved in the pathophysiology of sudden death after stroke [77]. 

Potentially, any ischemic lesion involving the central autonomic network (CAN) [78] and its pathways may lead to cardiovascular autonomic derangement. Insular and prefrontal cortex as well as the amygdala represent the highest control level of the CAN, resulting in a high-order processing of viscerosensory information and induction of an integrated autonomic feedback. Consequently, studies evaluating autonomic function in ischemic stroke have demonstrated an evident impact on cardiovascular control of insular cortex strokes [79,80]. 

Physiological studies on humans, examining the effect of either stimulation or inactivation of the two hemispheres, suggest a crucial role of the right hemisphere in establishing sympathetic tone, and of the left hemisphere in determining parasympathetic tone [81,82]. A higher prevalence of autonomic manifestations in right hemispheric strokes, compared to left strokes, has been suggested [83,84,85,86], but not confirmed by other authors [87,88,89]. Complex dysfunction of the cardiovascular autonomic regulatory system, with a significant reduction in parasympathetic modulation and an increase in sympathetic influence, has been likewise demonstrated in cases of bulbar ischemic stroke [90,91]. Therefore, the topography of ischemic lesions has a critical importance in determining the extent of autonomic dysfunction following stroke, yet the different etiologies of ischemic stroke may have a role as well. The classic TOAST classification of stroke etiologies denotes five subtypes of ischemic stroke: stroke due to either large-artery atherosclerosis (LAA), cardioembolism or small-vessel occlusion (lacunar infarction, LAC), and stroke of other determined etiology or of undetermined etiology [92]. Clear differences in HRV between patients with LAA stroke and LAC were observed [93,94], with LAA patients showing lower parasympathetic activity and higher sympathetic activity than the LAC group. Such divergence between LAA and LAC patients, if confirmed, may be due to the different size of the ischemic lesion, typically larger in LAA and more frequently involving the insular cortex, these two being among the main predictors of autonomic dysfunction after ischemic stroke [95], but possibly also to different risk factors underlying the different etiologies of stroke. Unfortunately, comparative studies between strokes of different etiologies generally exclude cardioembolic or cryptogenic stroke, partially as a result of technical issues in patients with episodes AF, such as the irregular fluctuation of ventricular response interval, that preclude the assessment by standard HRV analysis. However, in some cases, the number of patients involved is too small to draw conclusions [96].

Other possible predictors of ANS dysfunction are represented by the severity of stroke, as expressed by the National Institutes of Health Stroke Scale (NIHSS) [97,98], and coherently the size of the ischemic lesion, as well as premorbid conditions, including hypertension [99].

It is not clear whether the autonomic derangement visible in the acute phase of AIS can be completely reversible. Korpelainen et al. suggested that, within 6 months after AIS, the circadian rhythm of HRV may be restored [88], but the global 24 h HRV is persistently reduced [89]. Other authors confirm that HRV reduction may last up to 1 month [83] and 6 months [100] and, in a cohort of elderly stroke survivors who had otherwise made a good cognitive and physical recovery from stroke, for an average follow-up time of 9 months [101]. Dutsch et al. demonstrated the persistence of ANS derangement after a median time of 31 months following stroke [102]. Thus, impaired autonomic function may increase the risk of all-cause and cardiovascular mortality in stroke survivors for years after the ischemic event [103] and may itself have a role in enhancing the risk of stroke recurrence—for instance, by inducing the development of AF after AIS by neurogenic mechanisms [104].

### 2.3. The Predictive Role of HRV

The most recent research has shown that the analysis of HRV is a useful tool not only in risk stratification for stroke onset but also as a marker of post-stroke long-term outcomes. 

Abnormal HRV in the acute phase of stroke has been shown to predict unfavorable outcomes in the first 3 months after the ischemic event [105,106]. These findings were confirmed by our group [107], showing that loss of sympathetic rhythmic oscillation could predict a poorer 3-month outcome, and by autonomic evaluation [108,109]. HRV analysis may also predict acute complications of AIS, such as stroke-in-evolution, i.e., early worsening of neurological conditions [110], infarct expansion [95], infections within the first week [111] or possibly even sudden in-hospital death [86]. Furthermore, different papers suggest a possible correlation of reduced HRV with mortality at 1 month [112], at 1 year [95,113] and after a mean follow-up 7 years [114], respectively. Moreover, impairment in autonomic control could exert a negative influence on the capability of stroke survivors to comply with rehabilitation programs; coherently, HRV abnormalities are able to predict worse functional outcomes after rehabilitation, as suggested by Bassi et al. [115]. However, after adjustment for all demographic and clinical factors, Bassi et al. noted that post-stroke ANS dysfunction had a significant negative impact only in men. In particular, a decreased HRV was an independent predictor of rehabilitation outcome in men, but not in women [116]. Similarly, Arad et al. showed a correlation between HRV and functional performance, estimated with the functional independence measure (FIM) [117]. Furthermore, Guan et al. reported a higher risk of ischemic recurrence patients with precedent TIA or minor stroke showing changes in HF band power [118]. 

As we have previously described, ANS dysfunction may increase the risk of AIS; long-term autonomic derangement due to AIS could itself be a cofactor of the risk of recurrence of cerebrovascular events in stroke survivors. In the paper by Nayani et al., patients with ANS dysfunction had greater stroke severity at discharge, and more frequent cerebrovascular recurrences at 3-month follow-up, compared with patients without ANS dysfunction [95]. To conclude, assessment of cardiovascular autonomic function by means of HRV in the acute phase of ischemic stroke might yield relevant prognostic factors regarding acute complications, mortality and long-term outcomes, enabling the identification of high-risk patients who may benefit from specific secondary preventive measures.

## 3. Targeting the Vagus Nerve: A Promising Multilevel Approach

The molecular mechanisms and structural changes that involve the brain ischemic tissue have been extensively described over the years. Due to the rapid progression of the ischemic injury cascade, patients with AIS require immediate medical attention and urgent care. Thus, pharmacological (intravenous thrombolysis (IVT)) and mechanical (thrombectomy) treatments are recommended for selected patients affected by AIS and should be started as soon as possible. Recombinant tissue plasminogen activator (rt-PA) must be administrated within 4.5 h from the onset time of stroke symptoms, whilst surgical interventions should be performed within the first 6 h. However, less than one third of patients are eligible for these treatments [119,120,121]. Due to restrictions in use and possible complications (e.g., increased the risk of fatal intracranial hemorrhage), both interventions remain far from ideal and the one-month case-fatality rate remains high, with values ranging from 9 to 19% [122]. Moreover, the recanalization process is not without consequences. The re-intake of oxygen and glucose into the ischemic brain leads to an excessive production of reactive oxygen species that overwhelms endogenous antioxidant reserves and aggravates the tissue damage and the inflammatory response [123]. On the basis of these premises, the most recent research has focused on the need to operate through a multilevel approach based on two main points: neuroprotection during the acute phase and post-stroke neuroplasticity enhancement. Several treatments, mainly pharmaceuticals, which target one or more factors involved in the cascade of ischemia/reperfusion injury have been tested but failed or had small results [124]. 

ANS is extensively involved in the initiation, progression and outcome of stroke and, starting from the study of this intimate connection, it was possible to draw the basis for new therapeutic approaches. In particular, several pathophysiological aspects of stroke (e.g., neuroinflammation) are influenced by the activity of the vagus nerve [125]. In addition to depression and epilepsy, vagal stimulators have been already used in other chronic conditions such as chronic pain, refractory migraine and cardiovascular diseases, testifying to their safety profile [126,127]. Moreover, recent preclinical research on the cholinergic anti-inflammatory pathway is expanding the use of vagus nerve stimulation (VNS) to target neuroinflammation in neurological disorders such as Alzheimer’s disease, Parkinson’s disease and traumatic brain injuries [128,129,130]. Promising results are also shown in ischemic stroke models. 

### 3.1. The VNS Molecular Targets for Neuroprotection and Neuroplasticity in Stroke

The putative role of VNS goes beyond the function of re-establishing sympatho-vagal balance in cardiovascular autonomic control, lost following AIS, as seen in the previous chapter [131,132]. It is known that the afferent inputs triggered by invasive and non-invasive VNS activate and upregulate the α7 nicotinic acetylcholine receptors (α7nAchRs) which are expressed in astrocytes, microglia, endothelial cells and neurons [133,134,135]. Lu et al. proved that inhibiting α7nAchR expression prevents the expected improvement in rats subjected to VNS after the ischemic procedure, whereas the administration of a pharmacological α7nAchR agonist resulted in similar VNS-induced effects [136]. Even in the case of treatment with transcutaneous-auricular VNS (tVNS), improvements in neurofunctional tests were inhibited by receptor blockade [135]. The neuroprotective action deriving from the activation of α7nAchR seems to be the result of anti-inflammatory, antioxidant and antiapoptotic mechanisms (Figure 3). 

#### 3.1.1. Anti-Inflammatory Effects

Neuroinflammation plays a major role in the pathophysiology of stroke and represents one of the major targets for the treatment of stroke in current clinical and experimental research. The strong pro-inflammatory stimuli deriving from ischemic tissue induce the activation of microglia, astrocytes and endothelial cells, which start to produce pro-inflammatory cytokines. Leukocytes are recruited from the vessels and infiltrate the tissue by extravasation. Thus, cytotoxic substances are produced in large quantities (e.g., metalloproteases) and oxidative stress is aggravated [137,138]. Therefore, the suppression of post-stroke inflammation as a therapeutic approach can create the conditions for a better success of recanalization interventions and represents an alternative treatment for patients who do not qualify for thrombolysis. However, the pharmacological modifications of the inflammatory response must be carefully evaluated in order not to compromise the processes of tissue repair (e.g., secretion of growth factor) that are activated in parallel. Through the VNS, it is possible to recruit and enhance the cholinergic anti-inflammatory pathway by means of endogenous mechanisms. VNS has been shown to inhibit the expression levels of pro-inflammatory cytokines tumor necrosis factor-α (TNF-α), interleukin-6 (IL-6) and interleukin-1β (IL-1β) in the ischemic area [134,139,140]. The molecular mechanisms underlying this effect have yet to be well understood; however, some possible pathways have been identified. Lu et al. highlighted the involvement of phosphorylated Janus kinase 2 (p-JAK2)/phosphorylated signal transducer and activator of transcription 3 (p-STAT3) signal transduction pathway [136]. The intracellular signaling cascade determines the dimer p-STAT3’s translocation into the nucleus and the consequent expression of specific anti-inflammatory response factors, such as IL-10, which in turn promotes the silencing of pro-inflammatory genes [141,142,143]. In addition, a second regulatory mechanism operated by STAT3 has also been proposed, as it seems to interfere with the DNA binding of nuclear factor kappa-light-chain-enhancer of activated B cells (NF-kB), thereby reducing the production of pro-inflammatory cytokines [136,144]. The JAK2/STAT3 pathway’s activation by α7nAchR has also indirect effects aimed at restoring homeostasis and promoting repair processes. The in vitro study of Zhang et al. showed that the activation of α7nAchR through acetylcholine inhibits the transformation of resting microglia into the pro-inflammatory M1 phenotype and promotes the induction and maintenance of the anti-inflammatory M2 phenotype [145]. M2 microglia prolong neuron survival, phagocytosis and clearance of debris; hence, the selective modulation of M2 polarization has been proposed per se as a possible therapeutic strategy in acute stroke [145,146]. In practice, it was observed in a murine model of ischemic stroke that preventive treatment with tcVNS is able to attenuate ischemic and reperfusion injury by promoting polarization of the microglia to M2 [147]. Finally, peroxisome proliferator-activated receptors-γ (PPAR- γ) may have a role in the process of neuroinflammatory suppression, since PPAR- γ mRNA and protein levels were increased after brain ischemia and both were further upregulated after VNS treatment in an animal model [139]. Moreover, the VNS-mediated improvement was abolished and IL-1β and TNF-α levels were enhanced in an ischemic penumbra of rats with PPAR- γ silencing. 

#### 3.1.2. Antioxidant Effects {XE “3.2.2. Antioxidant Effects”}

The recruitment of the same anti-inflammatory cholinergic pathway seems to have antioxidant effects, as well. As a matter of fact, VNS significantly counteracted the oxidative process through the regulation of glutathione and superoxide dismutase levels in a rat model of focal cerebral ischemia and reperfusion [148]. α7nAchR intracellular signaling was demonstrated to be associated with the expression of antioxidant genes superoxide dismutase 1 (SOD1) and glutathione peroxidase 1 (GPX1) and the downregulation of pro-oxidative stress protein NADPH oxidase [149]. The JAK/PI3K/Akt and JAK2/STAT3/NF-kB pathways have been proposed as intracellular mechanisms to regulate this function. According to Parada et al., activation of the α7nAChR and JAK2/PI3K/Akt pathways ultimately leads to the release of nuclear factor erythroid 2-related factor 2 (NRF2) from Keap1. NRF2 migrates to the nucleus and binds to antioxidant response element (ARE) [150]. In addition to the upregulation of antioxidant genes, the reduction in ischemic insult could be due to the mitigation of reactive oxygen species (ROS) production by inhibition of inducible nitric oxide synthase (iNOS). iNOS is among the genes that are responsive to NF-kB, which is regulated by α7nAChR/JAK2 activation, as previously reported [151]. VNS-activated intracellular pathways seem to mediate not only transcriptional processes but also translation efficiency through microRNAs, as observed by Jiang et al. [152]. Although the specific molecular pathways that cause the decrease in ROS are not clear, it appears that microRNA 210 mediates mitochondrial ROS production [153]. 

#### 3.1.3. Antiapoptotic Effects {XE “3.2.3. Antiapoptotic Effects”}

Recent studies suggest that in addition to limiting the damage caused by inflammation and oxidative stress, VNS exerts direct antiapoptotic effects. The protein levels of both α7nAChR and p-Akt were significantly higher and the expression of cleaved caspase-3 was significantly less in the ischemic tissue of the VNS-treated group than in the untreated group [134]. Furthermore, VNS determined the downregulation of Bax and caspase-3 as well as the upregulation of Bcl-2, probably due to the inhibition of apoptotic cascades by lipocalin prostaglandin D2 Synthase (L-PGDS) [154]. Little is known about this enzyme in apoptosis suppression; however, it has been shown to play an important role in cell death and also in spinal cord injury, multiple sclerosis and Alzheimer’s disease [155].

#### 3.1.4. Neuroplasticity and Neurogenesis {XE “3.2.3. Neuroplasticity and Neurogenesis”}

In addition to the recruitment of mechanisms for neuroprotection in the acute phase of stroke, VNS has been shown to restore neuronal function, enhancing neuroplasticity and stimulating neurogenesis. Mounting evidence from studies has demonstrated that VNS, especially the non-invasive type, boosts axonal plasticity and reorganization of synaptic connectivity [135,156]. In Meyers’s retrograde transsynaptic tracing study, VNS resulted in a six-fold increase in synaptic connectivity in the lesioned hemisphere compared to equivalent rehabilitation without stimulation. Moreover, these modifications proved to be robust and enduring [157]. VNS drives the activation of locus coeruleus and nucleus basalis, thus engaging both the cholinergic and noradrenergic neuromodulatory systems and resulting in facilitation of the plasticity processes [158]. Even if the exact molecular steps remain to be clarified, recent research has uncovered a cholinergic-mediate non-neuronal transmission participating in the induction of long-term potentiation: an emerging view indicates that potentiation of cholinergic signaling increases interneuron firing rates either directly or via stimulation of astrocytes directly through the activation of α7nAChR [159,160,161]. The importance of temporal pairing between VNS and rehabilitative exercises further supports the hypothesis of an enhanced plasticity-dependent mechanism. The VNS-mediated release of neuromodulators during the performance of rehabilitative tasks enhances event-specific cortical plasticity [162,163]. Therefore, when stimulation is not associated with training, then it fails in recovering the lost function [162,164]. Finally, an inverted-U relationship between VNS-mediated plasticity and stimulation intensity has been highlighted, such that low and high intensities fail to drive plasticity and do not restore the affected functionality. The optimal intensity range for invasive VNS seems to lie between 0.6 and 1.0 mA in both preclinical and clinical stroke studies [165,166,167,168]. While it is logical to accept that stimulations at low intensities are not sufficient to reach the activation threshold of neuromodulation pathways [169,170], the neuronal mechanisms that limit plasticity at high intensities are still unclear. Two hypotheses have been formulated. First, the failure of higher intensity stimulation to enhance plasticity could be due to an overactivation and consequent desensitization phenomenon linked to G protein-coupled receptors for neuromodulators [171]. Alternatively, the inverted-U relationship may derive from differential activations of noradrenergic receptors. Low intensities of VNS would lead to the activation of high-affinity α-adrenergic receptors that promote synaptic plasticity. Meanwhile, high intensities of VNS would increase norepinephrine concentrations, promoting the recruitment of low-affinity β-adrenergic receptors and resulting in a pro-stability state [156,167,168,172]. Lastly, data suggest that activation of α7nAChR through invasive and non-invasive VNS upregulates the expression of neurotrophic and pro-angiogenic factors, such as brain-derived neurotrophic factor (BDNF), vascular endothelial growth factor (VEGF) and cerebral growth differentiation factor 11 (GDF11), in the ischemic penumbra [135,173,174,175]. However, to date, scanty data are available on the possible role of VNS in post-stroke neurogenesis and angiogenesis induction.

### 3.2. The Effects of VNS in Ischemic Stroke {XE “3.1. The Macro-Effects of VNS in Ischaemic Stroke”}

The efficacy of VNS has been demonstrated both in attenuating tissue damage, when performed in the acute phase of ischemic stroke, and in restoring lost functions, when applied during the chronic rehabilitation phase. VNS performed 30 min after reperfusion determined an approximately 50% reduction in infarct volumes compared to the untreated control group in all preclinical models of cerebral ischemia and reperfusion [134,140,176,177,178,179]. In addition, although a physiological post-stroke recovery was observed in control groups, the VNS treatment resulted in a more consistent and persistent improvement in functional scores, appreciable after only 24 h. Similar results were obtained with non-invasive VNS approaches. A single session of transcutaneous cervical vagus nerve stimulation (tcVNS) or tVNS, performed 30 min after the induction of ischemia, resulted in a significant difference in infarcted brain volume between the treated and the control groups and improved neurological functions [180,181,182]. In longer stimulation protocols (7 to 28 days), a continuous trend of neurofunctional restoration was observed in groups treated with tVNS [135,173,174,183]. 

Chronic impairment of the arm and hand is one of the most common consequences of stroke. Therefore, an additional effort in stroke research aims to develop new approaches targeting functional recovery that can be applicable and effective even long after the event. VNS has proved to be effective not only in the acute phase but also as a plasticity-enhancing therapy coupled with rehabilitation. In animal models, VNS performed during rehabilitative tasks restored pre-lesion successful rates, whereas training without VNS failed [157,184]. The recovery was long-lasting, as testified by the follow-up assessment at the first and seventh week after the cessation of VNS [157]. Further studies have shown that the therapeutic benefits of this approach strongly depend on the timing and amount of VNS. The best results were obtained when VNS was performed during rehabilitative training sessions on successful trials and this approach was found to be effective even when initiated several weeks after the ischemic event in subjects with chronic and stable deficits in motor function [185]. Delayed VNS, e.g., performed two hours after rehabilitative training sessions, resulted in significantly less improvement [162,185] and the increase in the number of stimulations also resulted in reduced recovery compared to a reduced number of stimulations paired with rehabilitative exercises [162]. Finally, Meyers et al. observed generalization of recovery to non-rehabilitated tasks: rats subjected to ischemic injury showed rapid functional recovery not only in the rehabilitation task associated with VNS, as described in previous studies, but also in exercises not belonging to the rehabilitation routine [157].

In humans, two randomized clinical trials confirmed the applicability of VNS paired with rehabilitation to improve limb function in chronic stroke patients with moderate to severe upper limb impairment [165,166]. Stable and long-lasting improvements were seen when in-clinic rehabilitation was followed by an individualized daily home training program paired with self-administered VNS, whereas researchers observed a parallel functional decline in the sham group (home exercises paired with stimulation at 0.0 mA). Notably, the participants assigned to sham stimulation experienced a similar benefit when they crossed over to active VNS treatment [166]. Moreover, 73% of participants demonstrated a clinically meaningful improvement, attested by an increase of more than 10 points in Upper Limb Fugl-Meyer Score, after 1-year follow-up of VNS paired with rehabilitation [186]. Recently, also the tVNS has been the subject of some clinical studies in the context of ischemic stroke. Capone and colleagues highlighted a priming effect of tVNS on subsequent robotic rehabilitative training of upper limb function in chronic stroke [187]. Patients were randomized to receive 1 h of real or sham tVNS immediately before a session of robotic therapy for 10 working days. After intervention, upper limb functionality was significantly better in the real-stimulation group as compared to the sham group. In two other studies, participants underwent 18 × 1-h rehabilitative sessions over 6 weeks, during which the tVNS was performed simultaneously with arm movements [188,189]. In addition to confirming the data on motor improvement, a recovery in tactile and proprioceptive sensory functions was also noticed. Furthermore, the efficacy rates of rehabilitation plus tVNS treatment seem to be comparable with the rates of invasive approach. As a matter of fact, in Redgrave et al.’s study, 83% of patients achieved a clinically meaningful response on the Upper Limb Fugl-Meyer Score [189]. Most participants felt that tVNS was comfortable, no one stopped the stimulation protocol due to tolerability problems, and there were no adverse events. 

The results of preclinical and clinical studies concerning non-invasive stimulation are particularly encouraging. However, while clinical trials on the effect of VNS in the chronic phase of stroke are numerous, to date, no studies on the application of VNS in humans during the acute phase of stroke are available. Since non-invasive VNS does not require surgical implantation and, given the proven safety of the treatment in the subacute phase, the recruitment and applicability of human studies with transcutaneous VNS should be less challenging than protocols that involve invasive VNS. Further research with similar populations, experimental design and timelines is needed to establish whether the efficacy of the two types of treatment is comparable. If so, non-invasive VNS could be introduced as an add-on treatment to the standard therapy in the acute phase of AIS, thus increasing the success rate of recanalization therapies and reducing reperfusion injury. Furthermore, considering the ease of its use and its possible beneficial effect in hemorrhagic stroke as well, non-invasive VNS could represent the very first therapeutic approach in the ultra-early management of stroke.

## 4. Conclusions

In the last few decades, researchers and clinicians have dedicated tremendous effort to reducing morbidity and mortality in ischemic stroke patients. However, the mortality rate remains considerably high and the chronic consequences of stroke impact severely the health-related quality of life of AIS survivors. 

There is a bidirectional loop between stroke and ANS, as the major risk factors for AIS are consequences of cardiovascular autonomic dysfunction and central nervous system lesions can lead to alterations in cardiovascular autonomic control. Moreover, ANS dysfunction following acute cerebrovascular events is strongly linked to poor prognosis in stroke patients. Therefore, cardiovascular autonomic control assessment by means of HRV analysis could have a key role in risk stratification for ischemic stroke and in prognosis of mid-term outcomes and clinical response to recanalization therapy. More recently, neuromodulation interventions, which aim to enhance the parasympathetic modulation of ANS, have emerged as a novel strategy for the management of stroke patients, both in the acute and chronic phases, due to their pleiotropic effects. In the latter area, little but promising data are available and an intriguing possibility for progress is offered by this new non-invasive stimulation methodologies.

## Figures and Tables

**Figure 1 ijms-22-02357-f001:**
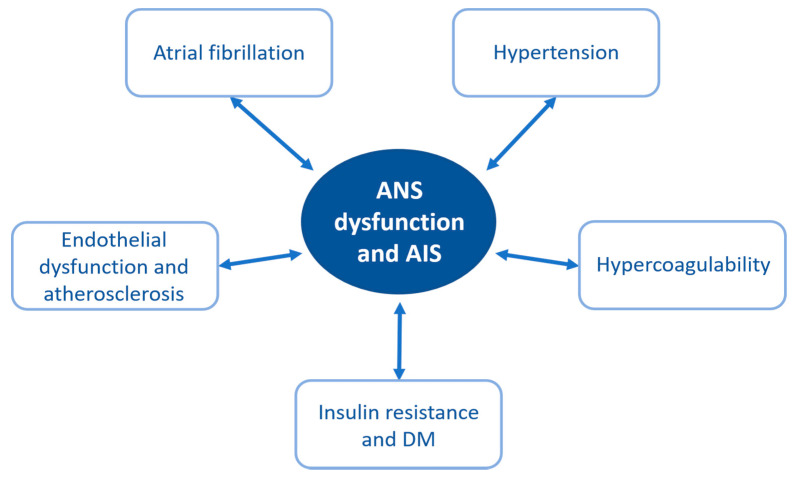
Risk factors for stroke and autonomic dysfunction. Abbreviations: ANS, autonomic nervous system; AIS, acute ischemic stroke; DM, diabetes mellitus.

**Figure 2 ijms-22-02357-f002:**
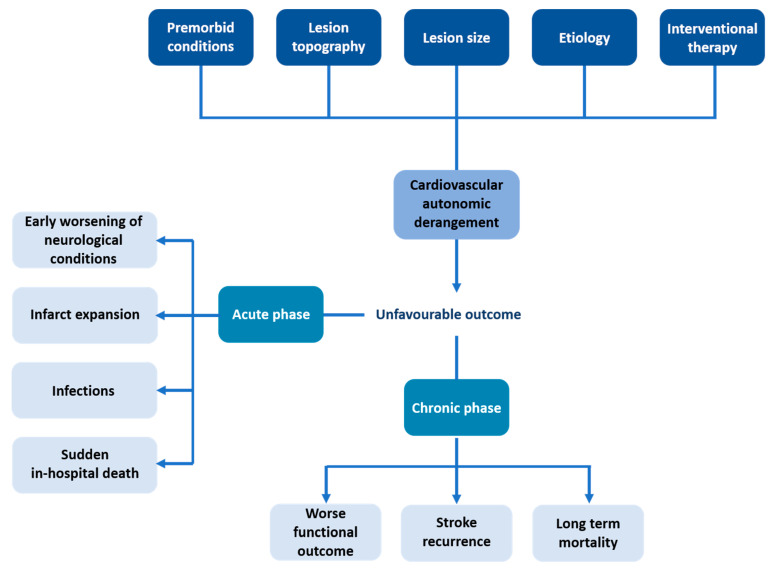
Cardiovascular autonomic derangement as a consequence of stroke and unfavorable outcomes in acute and chronic phase.

**Figure 3 ijms-22-02357-f003:**
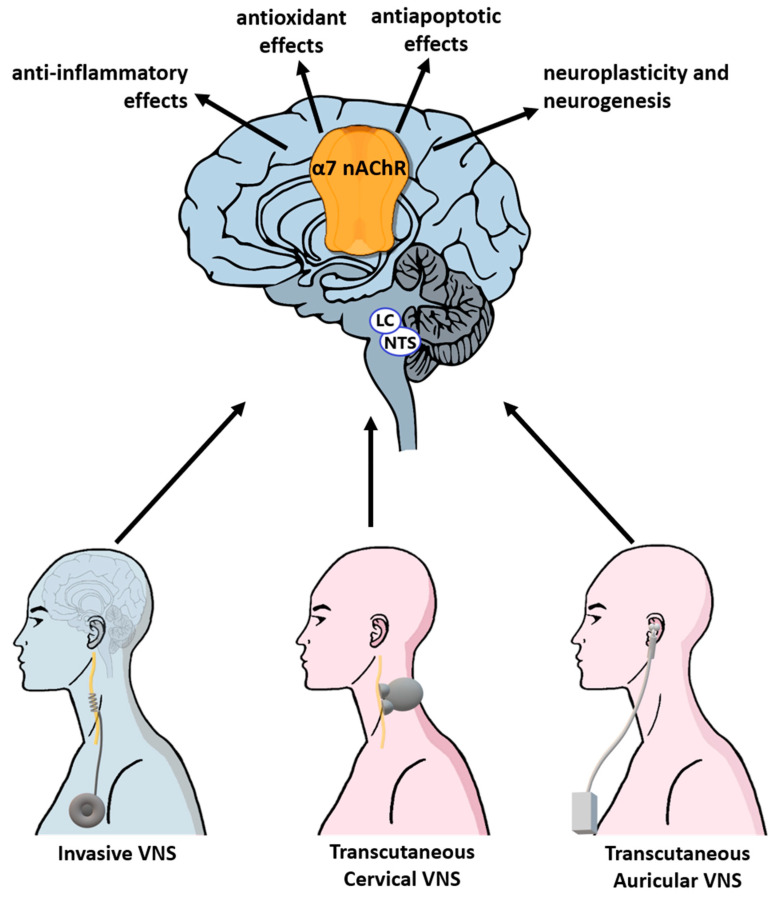
Central effects of invasive and non-invasive vagus nerve stimulation. Abbreviations: VNS, vagus nerve stimulation; LC, locus coeruleus; NTS, nucleus tractus solitarius; α7nAchR, α7 nicotinic acetylcholine receptor.

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
