# Peer review of "Targeting the Autonomic Nervous System for Risk Stratification, Outcome Prediction and Neuromodulation in Ischemic Stroke"

_ijms, 2021, doi:10.3390/ijms22052357_

Round 1

Reviewer 1 Report

This is a comprehensive review paper from the latest from theory principle to review of past researches to the latest update on clinical trials on the relationship between autonomic instability and post-stroke neuroplasticity and recovery. The manuscript is well-drafted and thorough. The readers will appreciate the clear review presented by the authors.

Author Response

We thank the Reviewer for her/his kind comment.

Reviewer 2 Report

The authors present an elegant review with the aim of estimating the emerging evidence linking the ANS to the pathological process of stroke, the effects of ANS dysregulation after the ischemic event, and the potential role of heart variability analysis in the diagnosis of stroke patients. They concluded that ANS dysfunction after an acute stroke is strongly related to poor prognosis and that neuromodulation  interventions are emerging as a new strategy for the management of stroke patients  both in acute and chronic phases due to pleiotropic effects. The study is potentially  interesting, but can be improved if the following considerations are addressed:

1.      It  would be interesting to include a comment that cardioembolic stroke is the subtype of ischemic infarction with the highest in-hospital mortality. The short-term prognosis of patients with cardioembolic stroke is poor compared to other ischemic stroke subtypes.
2.      The authors should include a comment regarding the fact that, in cardioembolic stroke, early recurrent embolization  is the most important  predictor for in- hospital mortality.
3.      The authors should analyze whether gender is a factor related to ANS dysfunction after acute stroke. It would be useful to mention in the text that  women differ from men in the distribution of risk factors and stroke subtype, stroke severity, and outcome.
4.     It would be helpful to mention that ischemic stroke may be the initial manifestation of hematologic diseases and also that hematological disorders are a commonly unrecognized cause of ischemic stroke. ANS dysfunction in cerebral ischemia due to hematological diseases is a clinical aspect that has not been assessed and that could also be of prognostic and outcome interest. This is another remarkable aspect that should be highlighted.

Author Response

We thank the Reviewer for her/his comment and suggestions. 

Reviewer 2

The authors present an elegant review with the aim of estimating the emerging evidence linking the ANS to the pathological process of stroke, the effects of ANS dysregulation after the ischemic event, and the potential role of heart variability analysis in the diagnosis of stroke patients. They concluded that ANS dysfunction after an acute stroke is strongly related to poor prognosis and that neuromodulation  interventions are emerging as a new strategy for the management of stroke patients  both in acute and chronic phases due to pleiotropic effects. The study is potentially  interesting, but can be improved if the following considerations are addressed:

  1. It would be interesting to include a comment that cardioembolic stroke is the subtype of ischemic infarction with the highest in-hospital mortality. The short-term prognosis of patients with cardioembolic stroke is poor compared to other ischemic stroke subtypes.

We thank the Reviewer for this comment. We added data on cardioembolic stroke in our revised manuscript (lines 108-116).

  1. The authors should include a comment regarding the fact that, in cardioembolic stroke, early recurrent embolization is the most important  predictor for in- hospital mortality.

We thank the Reviewer for this suggestion. We added a comment on early embolic recurrence in our revised manuscript (lines 113-116).

  1. The authors should analyze whether gender is a factor related to ANS dysfunction after acute stroke. It would be useful to mention in the text that women differ from men in the distribution of risk factors and stroke subtype, stroke severity, and outcome.

We thank the Reviewer for her/his observation and suggestion. Several epidemiological studies have shown that stroke incidence is about 25% to 33% higher among men than women until advanced age. A higher incidence of stroke in women after 85 years have been also highlighted, due both to a longer life expectancy and a progressive post-menopausal loss of the neuroprotective role of estrogen (Girijala at al., 2017). Regarding sex differences in stroke subtypes, the age-adjusted male-female incidence rate ratio is 1.55 (95% CI, 1.48 to 1.61) for ischemic stroke, 1.60 (95% CI, 1.47 to 1.74) for intracerebral hemorrhage, 0.84 (95% CI, 0.69 to 1.04) for subarachnoid hemorrhage, and 1.08 (95% CI, 0.98 to 1.20) for stroke of undetermined cause (Appelros et al., 2009). Unfortunately, few studies have given age specific data on men and women separately for ischemic subtypes. Moreover, women experience more severe strokes than men and have longer hospitalizations. Therefore, the percentages of permanent disability and mortality are also higher in women (Appelros et al., 2010; Bushnell et al., 2018).

The majority of stroke risk factors, such as those mentioned in our manuscript (e.g., hypertension, AF and diabetes mellitus) are more common in males, but when considering only stroke patients, they are shown to increase stroke risk more in females (Girijala at al., 2017).

We have added the above epidemiological information to the manuscript (lines 41-44, 70-72).

One of the high impact stroke-related outcome in women is the post-stroke depression, probably related to the greater social isolation experienced by women who have suffered a stroke (Poynter et al., 2009; Volz et al., 2019). However, this condition further worsens the quality of life and contributes to increase mortality and decrease functional recovery.

Sex-related differences in cardiovascular autonomic control have generally been reported in the normal population and HRV studies suggest a larger vagal modulation in women, despite they are characterized by a higher HR (Koenig & Thayer, 2016). As regards the alterations of ANS after acute stroke, to the best of our knowledge, no studies showed significant sex differences but the topic is little investigated and should be deepened with ad hoc studies. However, as reported in the first version of our manuscript, after the adjustment for all demographic and clinical factors, Bassi et al. noted that ANS dysfunction had a significant negative impact only in men. In particular, a decreased HRV was an independent predictor of rehabilitation outcome in men, but not in women (Bassi et al., 2010). We better described these observations in the new version of our manuscript (lines 269-273).

  1. It would be helpful to mention that ischemic stroke may be the initial manifestation of hematologic diseases and also that hematological disorders are a commonly unrecognized cause of ischemic stroke. ANS dysfunction in cerebral ischemia due to hematological diseases is a clinical aspect that has not been assessed and that could also be of prognostic and outcome interest. This is another remarkable aspect that should be highlighted.

We thank the Reviewer for this useful comment. As suggested, a synthetic overview on thrombophilic and other hematological disorders that are associated to an increased risk of stroke has been added (lines 165-182), with particular reference to their possible link to ANS.

Reviewer 3 Report

Ref: ijms-1116086

Title: Targeting the Autonomic Nervous System for Risk Stratification, Outcomes Prediction and Neuromodulation in Stroke

Recommendation: Accept for publication

Very well written review. Nicely described and informative schemes are an additional value. The topic of the review is novel and will certainly find many readers.

Author Response

We thank the Reviewer for this kind comment.